# Fast convergence of stochastic subgradient method under interpolation

**Huang Fang, Zhenan Fan & Michael P. Friedlander**
Department of Computer Science
University of British Columbia
Vancouver, BC, Canada
{hgfang, zhenanf, mpf}@cs.ubc.ca

## ABSTRACT

This paper studies the behaviour of the stochastic subgradient descent (SSGD) method applied to over-parameterized nonsmooth optimization problems that satisfy an interpolation condition. By leveraging the composite structure of the empirical risk minimization problems, we prove that SSGD converges, respectively, with rates $\mathcal{O}(1/\epsilon)$ and $\mathcal{O}(\log(1/\epsilon))$ for convex and strongly-convex objectives when interpolation holds. These rates coincide with established rates for the stochastic gradient descent (SGD) method applied to smooth problems that also satisfy an interpolation condition. Our analysis provides a partial explanation for the empirical observation that sometimes SGD and SSGD behave similarly for training smooth and nonsmooth machine learning models. We also prove that the rate $\mathcal{O}(1/\epsilon)$ is optimal for the subgradient method in the convex and interpolation setting.

## 1 INTRODUCTION

Gradient descent (GD) and subgradient descent (subGD) methods are simple and effective first-order optimization algorithms for training machine learning models. The convergence-rate analyses for these methods depend crucially on the smoothness of the objective function. It is well understood that there is a fundamental gap between the convergence rates of gradient-like methods for smooth and nonsmooth problems (Shor, 1984; Nemirovski & Yudin, 1983; Nesterov, 2005; Bubeck, 2015; Beck, 2017). Table 1 summarizes the rates for the main problem classes.

However, in the practice of training machine learning models, the nonsmootheness from the model is not causing much trouble (Glorot et al., 2011; Goodfellow et al., 2016). Neural networks with nonsmooth activation function such as ReLU activation can usually be trained as fast as the one with smooth activation function such as softplus, see Figure 1.1 as a motivating experiment from us to compare the convergence of gradient and subgradient based methods for smooth and nonsmooth neural networks on the MNIST dataset[1] to classify digits 0 and 1. We can see that the gradient and subgradient based methods, either batch or stochastic version, has similar convergence behaviour for smooth and nonsmooth neural networks. Therefore there is a discrepancy between theory and practice.

The success of overparameterized models—such as deep and wide neural networks—has instigated a trend in the analysis of stochastic variants of gradient descent in the *interpolation* setting, in which the model achieves zero training loss, and therefore fits all of the training data (Schmidt & Le Roux, 2013; Bassily et al., 2018; Ma et al., 2018b; Jain et al., 2018; Vaswani et al., 2019a;b; Wu et al., 2019; Liu & Belkin, 2020). This recent series of papers offer insight into the fast convergence of SGD and new approaches for algorithm design. This line of analysis, however, focus exclusively on smooth objective function, and cannot explain the effectiveness of SSGD for training nonsmooth neural networks.

We present a formal analysis showing that SSGD for nonsmooth objectives could converge as fast as smooth objectives in the interpolation setting. Our contributions include:

---

[1] http://yann.lecun.com/exdb/mnist/

| objective $f(\mathbf{x})$ | smooth | nonsmooth |
|---|---|---|
| convex | $\mathcal{O}(1/\epsilon)$ | $\mathcal{O}(1/\epsilon^2)$ |
| strongly convex | $\mathcal{O}(\log(1/\epsilon))$ | $\mathcal{O}(1/\epsilon)$ |

Table 1: Worst-case iteration complexity of batch gradient and subgradient methods.

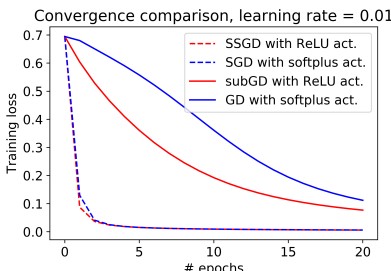

Figure 1.1: The convergence of gradient-based (GD and SGD) and subgradient-based (subGD and SSGD) methods for smooth and nonsmooth neural networks

- A description of semi-smoothness properties of a function useful for the iteration complexity analysis of convex objectives in the interpolation context. Under mild conditions, semi-smoothness under interpolation allows us to prove that SSGD has iteration complexity $\mathcal{O}(1/\epsilon)$ for convex objectives, and $\mathcal{O}(\log(1/\epsilon))$ for strongly-convex objectives. These rates improved the classic bounds $\mathcal{O}(1/\epsilon^2)$ and $\tilde{\mathcal{O}}(1/\epsilon)$ for convex and strongly-convex objectives and match the convergence rates of SGD for convex and smooth objective under interpolation.

- Proof that the iteration bound $\mathcal{O}(1/\epsilon)$ is optimal in the convex and interpolation setting. In contrast to the case with a smooth objective function, subgradient-based methods cannot be further accelerated for nonsmooth model—even with the interpolation assumption.

## 2 RELATED WORK

### 2.1 FIRST-ORDER METHODS FOR NONSMOOTH OPTIMIZATION

The subgradient method and its convergence analysis for general convex and nonsmooth problems was first described by Shor in the late 1960s and 70s; see Shor (1984). Nemirovski & Yudin (1983) subsequently established that the iteration complexity $\mathcal{O}(1/\epsilon^2)$ described by Shor (1984) is optimal for methods that can only access a subgradient oracle. Subsequent works for minimizing general convex and nonsmooth problems include stochastic subgradient methods (Polyak, 1987; Shalev-Shwartz et al., 2007; Shamir & Zhang, 2013), dual averaging (Nesterov, 2009), and acceleration via smoothing (Nesterov, 2005; Beck & Teboulle, 2012). More recently, Zhang et al. (2020) and Shamir (2020), among others, described the tractability and complexity of getting an approximate stationary point for general nonsmooth and nonconvex problems.

A related line of work involves the analysis of optimization algorithms for partially nonsmooth objectives that take the form $f(\mathbf{x}) + g(\mathbf{x})$, where the function $f$ is smooth, and the function $g$, which usually represents a regularizer, is convex and nonsmooth. Many models in feature selection and compressed sensing fall into this category of problems, which are usually solved by variations of the proximal-gradient method (Nesterov, 2007; Beck & Teboulle, 2009; Xiao, 2010; Parikh & Boyd, 2014; Defazio et al., 2014; Allen-Zhu, 2017). These approaches typically use special properties of the regularization function $g$, which generally do not apply in the context we consider in this paper.

Another related line of works focus on compositional optimization (Drusvyatskiy & Paquette, 2019; Davis & Drusvyatskiy, 2018; Duchi & Ruan, 2018). These works consider the objective as the compositions of convex functions and smooth maps, which is different from the objective formulation in this work, see section 3 for details.

### 2.2 INTERPOLATION HELPS OPTIMIZATION

The interpolation condition implies that the residual between the prediction of the model and data vanishes. In the context of nonlinear least-squares, for example, it is known that interpolation

---

**Algorithm 1** Stochastic subgradient descent. The learning rate function $\alpha_t : \mathbb{N} \to \mathbb{R}_+$ returns the learning rate at iteration $t$.

---

1: **Initialize:** $w^{(1)} \in \mathbb{R}^d$
2: **for** $t = 1, 2, \ldots$ **do**
3:     select $i \in \{1, 2, \ldots, n\}$ uniformly at random
4:     compute $g^{(t)} \in \partial f_i(w^{(t)})$
5:     $w^{(t+1)} = w^{(t)} - \alpha_t g^{(t)}$
6: **end for**

---

allows the Gauss-Newton method to converge at a local superlinear rate, although in general it can only achieve a local linear rate (Bertsekas, 1997). Conditional gradient method can also enjoy a linear convergence rate for least square objective when interpolation holds (Beck & Teboulle, 2004). Recent interest in over-parameterized models for machine learning problems makes interpolation (or near interpolation) a relevant assumption for the analysis and development of new algorithms. Under interpolation, stochastic gradient descent enjoys the same convergence rate as its deterministic counterpart, gradient descent (Schmidt & Le Roux, 2013; Ma et al., 2018a; Bassily et al., 2018). Some techniques that originally developed for gradient descent, such as line-search and Nesterov acceleration, are proven to also hold for SGD when interpolation is satisfied (Vaswani et al., 2019b;a; Liu & Belkin, 2020; Jain et al., 2018) Related work in this vein includes improved analysis of stochastic second-order optimization algorithms (Meng et al., 2020) and adaptive stochastic gradient descent (Loizou et al., 2020; Vaswani et al., 2020) under interpolation. Another line of work (Qian et al., 2019; Khaled & Richtárik, 2020) in this direction aims to explore assumptions other than the classic bounded noise assumption for the optimization theory of training over-parameterized model, such as the expected growth condition (Schmidt & Le Roux, 2013), expected smoothness (Qian et al., 2019; Gower et al., 2018), etc. All of the work just mentioned involves differentiable and smooth objective functions, and cannot explain the fast convergence of SSGD for nonsmooth objectives that is often observed in practice.

## 3 PROBLEM SETTING

We consider the unconstrained empirical risk-minimization problem

$$\underset{w \in \mathbb{R}^d}{\text{minimize}} \quad f(w) := \frac{1}{n} \sum_{i=1}^{n} f_i(w) \qquad \text{where each} \qquad f_i(w) := \ell(h_i(w)) \tag{3.1}$$

and $n$ is the number of data points. Throughout the paper, we use $w^*$ to denote any solution of eq. (3.1), and thus $f^* := f(w^*)$ is the optimal objective value. We assume that the 1-dimensional loss function $\ell : \mathbb{R} \to \mathbb{R}_{\geq 0}$ is nonnegative, $\inf \ell = 0$, convex, and is 1-smooth, i.e.,

$$|\ell'(\alpha) - \ell'(\beta)| \leq |\alpha - \beta| \quad \forall \alpha, \beta \in \mathbb{R}.$$

Without loss of generality, we also assume $\ell(0) = 0$. Common examples for the loss function include the 2-norm, logistic loss, and the 2-norm hinge loss function. The $n$ functions $h_i$ are Lipschitz continuous with respect to the fixed parameter $L$:

$$|h_i(w_1) - h_i(w_2)| \leq L\|w_1 - w_2\| \quad \forall w_1, w_2 \in \mathbb{R}^d, \ \forall i \in [n].$$

We make no assumption on their smoothness properties. Many key machine learning tasks can be formulated as (3.1), including training deep neural networks with nonsmooth activations, such as the ReLU function. Here and throughout, the function $\| \cdot \|$ is the 2-norm of a vector, unless otherwise specified.

Our analysis relies on the Clark *generalized* gradient (Clarke, 1990) of the nonconvex and nonsmooth function $h$, defined as the convex hull of all valid limiting gradients:

$$\partial h(w) := \text{conv} \left\{ u \mid u = \lim_{k \to \infty} \nabla h(w_k), \ w_k \to w \right\},$$

This definition additionally requires $h$ to be almost everywhere differentiable by Rademacher's theorem. We refer readers to Clarke (1990), and more recently, to Zhang et al. (2020), who use this generalized gradient in a related analysis. These properties of the generalized gradient are needed for our analysis:

(chain rule) $\qquad\qquad \partial f_i(w) = \ell'(h(w)) \cdot \partial h(w),$

(gradient bound) $\qquad\quad \|\partial h_i(w)\| \leq L, \ \forall w \in \mathbb{R}^d, \ \forall i \in [n].$

The second property follows from the $L$-Lipschitz continuity of each function $h_i$. We define the norm of the generalized gradient at a vector $w$ as

$$\|\partial h_i(w)\| = \sup \{ \|z\| \mid z \in \partial h_i(w) \}.$$

Algorithm 1 describes the stochastic subgradient descent (SSGD) method. Interpolation means that our model has the ability to fit all training samples perfectly and therefore we can obtain zero training loss at the solution e.g., $f(w^*) = 0$. The interpolation assumption holds for overparameterized neural networks and is gaining increasing interest in recent years Jacot et al. (2018).

## 4 MAIN RESULTS

### 4.1 BOUNDS AND LIPSCHITZ PROPERTIES OF THE GENERALIZED GRADIENT

Our analysis hinges on establishing that the objective function $f$ is differentiable at all of its minimizers, i.e., at all vectors $w$ such that $f(w) = 0$. This is implied by the following proposition, which holds even without the interpolation condition.

---

**Proposition 4.1** (Generalized growth condition). *For all $d$-vectors $w$,*

$$\|\partial f_i(w)\|^2 \leq 2L^2 f_i(w) \qquad \forall i \in [n]. \tag{4.1}$$

*Consequently,*

$$\|\partial f(w)\|^2 \leq \frac{1}{n} \sum_{i=1}^{n} \|\partial f_i(w)\|^2 \leq 2L^2 f(w). \tag{4.2}$$

---

Equation (4.2) implies that if the objective value $f(w) = 0$, then the generalized gradient contains only the origin: $\partial f(w) = \{0\}$. It thus follows from Clarke (1981, Property 10) that $f$ is differentiable at any point with zero objective value. This means that if there is a solution with a zero value, then it must be a fixed point of the subgradient method. This property stands in contrast to a solution where the function is nonsmooth and makes it possible for the subgradient method to converge to solution with constant learning rate.

*Proof of Proposition 4.1.* Fix an arbitrary $d$-vector $w$. The subdifferential $\partial f_i(w) = \ell'(h_i(w)) \cdot \partial h_i(w)$. Because each function $h_i$ is $L$-Lipschitz continuous, we can then deduce that

$$
\begin{aligned}
\|\partial f_i(w)\|^2 &\leq L^2 \left[ \ell'(h_i(w)) \right]^2 \\
&\overset{(i)}{=} L^2 (\ell'(h_i(w)) - \ell'(0))^2 \\
&\overset{(ii)}{\leq} 2L^2 (\ell(h_i(w)) - \ell(0)) \\
&= 2L^2 f_i(w).
\end{aligned} \tag{4.3}
$$

Step (i) follows from the assumption that $\ell(0) = \min_{\lambda \in \mathbb{R}} \ell(\lambda) = 0$, which implies $\ell'(0) = 0$. Step (ii) follows from the fact that any convex $L$-smooth function $u : \mathbb{R}^n \to \mathbb{R}$ satisfies the bound

$$u(y) - u(x) - \langle \nabla u(x), y - x \rangle \geq \frac{1}{2L} \|\nabla u(y) - \nabla u(x)\|^2;$$

see Nesterov (2014, Theorem 2.1.5). Make the identifications

$$u = \ell, \quad y = h_i(w), \quad x = 0$$

to immediately obtain eq. (4.3) and thus the proof for eq. (4.1).

Equation (4.2) can be obtained directly from Jensen's inequality:

$$\|\partial f(w)\|^2 = \left\| \frac{1}{n} \sum_{i=1}^{n} \partial f_i(w) \right\|^2 \leq \frac{1}{n} \sum_{i=1}^{n} \|\partial f_i(w)\|^2 \leq \frac{1}{n} \sum_{i=1}^{n} 2L^2 f_i(w) = 2L^2 f(w).$$

$\qquad\qquad\qquad\qquad\qquad\qquad\qquad\qquad\qquad\qquad\qquad\qquad\qquad\qquad\qquad\qquad\qquad\quad \square$

The composite structure of the functions $f_i$ allows us develop a semi-Lipschitz bound on their generalized gradients. Moreover, we show that it is possible to provide a global convex majorant for each function $f_i$, which holds without assuming convexity. Interestingly, eq. (4.5) overlaps with the "semi-smoothness" property of over-parameterized neural networks derived by Allen-Zhu et al. (2019, Theorem 4). In contrast to that result, however, we do not use any special properties of over-parameterized neural networks. Proposition 4.2 may thus be of more general interest.

---

**Proposition 4.2** (Semi-smoothness). *For all vectors $w_1$ and $w_2$, and each $i \in [n]$,*

$$\|\partial f_i(w_2) - \partial f_i(w_1)\| \leq L^2 \|w_2 - w_1\| + 2L\sqrt{2\min\{f_i(w_1), f_i(w_2)\}}, \qquad (4.4)$$

*and*

$$f_i(w_2) \leq f_i(w_1) + \langle \partial f_i(w_1), w_2 - w_1 \rangle + \frac{L^2}{2}\|w_2 - w_1\|^2 + 2L\|w_2 - w_1\|\sqrt{2f_i(w_1)}. \tag{4.5}$$

---

See Appendix A for the proof of this result.

## 4.2 CONVERGENCE RATE OF STOCHASTIC SUBGRADIENT DESCENT

We now present a global convergence analysis for the SSGD algorithm under the additional assumption that the objective $f$ of eq. (3.1) is convex. We develop a bound on the expected progress of the objective value that depends on the minimizing value $f^*$, rather than on the Lipschitz bound on the function itself, which is the usual bound in the literature. Our proof is based on a simple modification of the classical proof of subgradient descent method.

---

**Theorem 4.3** (Global convergence rate of SSGD). *Assume $f$ is convex. Then for any positive integer $T$ and any learning rate function $\alpha_t$ that satisfies $\sum_{t=1}^{T}(\alpha_t - L^2\alpha_t^2) > 0$,*

$$\min_{t \in [T]} \mathbb{E}[f(w^{(t)}) - f^*] \leq \frac{\|w^{(1)} - w^*\|^2 + 2L^2 f^* \sum_{t=1}^{T} \alpha_t^2}{2\sum_{t=1}^{T}(\alpha_t - L^2\alpha_t^2)}. \tag{4.6}$$

---

*Proof.* Let $g_i^{(t)} \in \partial f_i(w^{(t)})$. Then each iterate $w^{(t)}$ satisfies the bound

$$\mathbb{E}[\|w^{(t+1)} - w^*\|^2 \mid w^{(t)}] = \frac{1}{n}\sum_{i=1}^{n}\|w^{(t)} - \alpha_t g_i^{(t)} - w^*\|^2$$

$$= \|w^{(t)} - w^*\|^2 - 2\alpha_t \left\langle \frac{1}{n}\sum_{i=1}^{n} g_i^{(t)}, w^{(t)} - w^* \right\rangle + \frac{1}{n}\sum_{i=1}^{n}\alpha_t^2\|g_i^{(t)}\|^2$$

$$\overset{(i)}{\leq} \|w^{(t)} - w^*\|^2 - 2\alpha_t(f(w^{(t)}) - f(w^*)) + \frac{1}{n}\sum_{i=1}^{n}\alpha_t^2\|g_i^{(t)}\|^2 \qquad (4.7)$$

$$\overset{(ii)}{\leq} \|w^{(t)} - w^*\|^2 - 2\alpha_t(f(w^{(t)}) - f(w^*)) + 2\alpha_t^2 L^2 f(w^{(t)}), \qquad (4.8)$$

where (i) follows from the convexity of $f$, and (ii) follows from Proposition 4.1. Take expectations on both sides of the inequality (4.8) and rearrange to obtain

$$(2\alpha_t - 2L^2\alpha_t^2) \cdot \mathbb{E}[(f(w^{(t)}) - f^*)] \leq \mathbb{E}[\|w^{(t)} - w^*\|^2] - \mathbb{E}[\|w^{(t+1)} - w^*\|^2] + 2L^2\alpha_t^2 f^*. \tag{4.9}$$

Summing inequality (4.9) over $t \in \{1, 2, \ldots, T\}$ yields

$$\sum_{t=1}^{T}(2\alpha_t - 2L^2\alpha_t^2)\mathbb{E}[f(w^{(t)}) - f^*] \leq \|w^{(1)} - w^*\|^2 + 2L^2 f^* \sum_{t=1}^{T}\alpha_t^2.$$

Divide both sides by $2\sum_{t=1}^{T}(\alpha_t - L^2\alpha_t^2) > 0$ to obtain the required result. $\qquad\square$

This proof mirrors closely the classical proof of SSGD, which assumes that the subdifferential of each $f_i$ is bounded, i.e., $\|\partial f_i(w)\| \leq G$ for some constant $G$. We use Proposition 4.1 in eq. (4.7) to avoid the bounded subgradient assumption and to express the convergence rate using the minimal value $f^*$. This modification allows us to leverage the interpolation assumption that $f^* = 0$. We use Theorem 4.3 to immediately deduce the following convergence rate result.

---

**Corollary 4.4** (Global convergence rate of SSGD with constant learning rate)**.** *Assume that $f$ is convex and that the learning rate $\alpha_t = 1/(2L^2)$ is constant for all $t > 0$. Then for any positive integer $T$, the SSGD iterates $w^t$ satisfy*

$$\min_{t \in [T]} \mathbb{E}[f(w^{(t)}) - f^*] \leq (2L^2/T)\|w^{(1)} - w^*\|^2 + f^*.$$

*Furthermore, when $f^* = 0$ (interpolation holds),*

$$\min_{t \in [T]} \mathbb{E}[f(w^{(t)})] - f^* \leq (2L^2/T)\|w^{(1)} - w^*\|^2.$$

---

Corollary 4.4 indicates that the SSGD method converges at rate $\mathcal{O}(1/\epsilon)$ when interpolation holds. This rate matches the convergence rate of SGD for smooth objective functions under interpolation (Schmidt & Le Roux, 2013). When interpolation does not hold, Corollary 4.4 implies that SSGD, on expectation, could obtain objective value lower than $2f^* + \epsilon$ in $\mathcal{O}(1/\epsilon)$ time, which could be close to $f^*$ when $f^*$ is close to 0 (interpolation nearly holds).

Next, we derive convergence rate under the stronger assumption that $f$ is strongly convex, which means that

$$f(w_1) \geq f(w_2) + \langle g, w_1 - w_2 \rangle + \frac{\mu}{2}\|w_1 - w_2\|^2 \quad \forall w_1, w_2 \in \mathbb{R}^d, \ \forall g \in \partial f(w_2) \qquad (4.10)$$

for some constant $\mu > 0$. Note that recent works indicated that in order to prove linear convergence rate, the strong convexity assumption can be relaxed to other weaker assumptions that could hold even for some nonconvex functions (Karimi et al., 2016; Qian et al., 2019). For simplicity, we assume strong convexity in our analysis.

---

**Theorem 4.5** (Global convergence rate of SSGD under strong convexity)**.** *Assume that $f$ is $\mu$-strongly convex and that the learning rate $\alpha_t = 1/L^2$ is constant for all $t > 0$. Then for any positive integer $T$, the SSGD iterates $w^{(t)}$ satisfy*

$$\mathbb{E}[\|w^{(T)} - w^*\|^2] \leq \left(1 - \frac{\mu}{L^2}\right)^{T-1} \|w^{(1)} - w^*\|^2 + \frac{2}{\mu}f^*. \qquad (4.11)$$

---

*Proof.* Use the definition of the SSGD iteration to obtain

$$\mathbb{E}[\|w^{(t+1)} - w^*\|^2 \mid w^{(t)}]$$

$$\overset{(i)}{=} \|w^{(t)} - w^*\|^2 - 2\alpha_t \left\langle \frac{1}{n}\sum_{i=1}^{n} g_i^{(t)}, w^{(t)} - w^* \right\rangle + \frac{1}{n}\sum_{i=1}^{n} \alpha_t^2 \|g_i^{(t)}\|^2$$

$$\overset{(ii)}{\leq} \|w^{(t)} - w^*\|^2 - 2\alpha_t(f(w^{(t)}) - f^*) - \mu\alpha_t\|w^* - w^{(t)}\|^2 + \frac{1}{n}\sum_{i=1}^{n} \alpha_t^2 \|g_i^{(t)}\|^2$$

$$\overset{(iii)}{\leq} (1 - \mu\alpha_t)\|w^{(t)} - w^*\|^2 - 2\alpha_t(f(w^{(t)}) - f^*) + 2\alpha_t^2 L^2 f(w^{(t)}) \qquad (4.12)$$

$$\overset{(iv)}{=} \left(1 - \frac{\mu}{L^2}\right)\|w^{(t)} - w^*\|^2 + \frac{2}{L^2}f^*, \qquad (4.13)$$

where (i) follows from the same argument of the proof of Theorem 4.3; (ii) follows from the $\mu$-strong convexity of $f$ (see eq. (4.11)); (iii) follows from eq. (4.1); and (iv) follows from the definition of the learning rate $\alpha_t = 1/L^2$.

Taking expectation to both sides of eq. (4.13) and recursively apply it to $t \in \{1, 2, \ldots, T\}$ to deduce

$$\mathbb{E}[\|w^{(T)} - w^*\|^2] \leq \left(1 - \frac{\mu}{L^2}\right)^{T-1} \|w^{(1)} - w^*\|^2 + \sum_{t=0}^{T-2} \left(1 - \frac{\mu}{L^2}\right)^t \frac{2}{L^2} f^*$$

$$\overset{\text{(i)}}{\leq} \left(1 - \frac{\mu}{L^2}\right)^{T-1} \|w^{(1)} - w^*\|^2 + \frac{2}{\mu} f^*, \tag{4.14}$$

where (i) follows from the fact that $\sum_{t=0}^{\infty} (1 - \frac{\mu}{L^2})^t = L^2/\mu$.

$\square$

Theorem 4.5 indicates that the SSGD can converges to the ball centered at $w^*$ with radius $2\sqrt{f^*/\mu}$ at a linear rate. If additionally interpolation holds, SSGD converges to the solution linearly. Again, this rate matches the convergence rate of SGD for smooth and strongly convex objectives in the interpolation setting (Schmidt & Le Roux, 2013; Ma et al., 2018a). Similar to Corollary 4.4, when interpolation is nearly satisfied, SSGD converges linearly to an $2\sqrt{f^*/\mu}$-approximate solution.

Corollary 4.4 and Theorem 4.5 also provide insight into the effect of learning rate schedules on the performance of SSGD. A learning rate schedule $\alpha_t$ that decays as $t^{-1/2}$ is optimal because it causes the algorithm to exhibit a complexity bound of $\mathcal{O}(1/\epsilon^2)$, which is the theoretical lower bound (Nemirovski & Yudin, 1983). However, the learning rate schedule $\mathcal{O}(t^{-1/2})$ is slow in practice. Corollary 4.4 and Theorem 4.5 partially explain the discrepancy between the theory and practice: many machine learning models exhibit interpolation or near interpolation, and an aggressive constant learning-rate schedule works better than the conservative worst-case optimal learning rate $\alpha_t = \mathcal{O}(t^{-1/2})$.

Other than the learning rate scheduling $\alpha_t = 1/L^2$, we can adopt a more carefully tuned learning rate scheduling (Stich, 2019) to obtain a refined convergence rate, see more details in Appendix.

### 4.3 LOWER BOUNDS

We have proven that, under interpolation, SGD for smooth problems and SSGD for nonsmooth problems exhibit the same convergence rates. This causes us to consider the following questions:

- Is it possible to induce momentum-type acceleration for SSGD under interpolation? Vaswani et al. (2019a) and Liu & Belkin (2020) showed recently that acceleration is possible for SGD under interpolation. It seems plausible that similar techniques could be used for nonsmooth problems.

- Can the composite structure $f_i = \ell \circ h_i$, which is central to our analysis, be used to establish an improved convergence rate for SSGD without interpolation? In other words, we know the lower bound on the iteration complexity for any subgradient method for nonsmooth functions is $\Omega(1/\epsilon^2)$. What, then, is the lower bound for minimizing the structured nonsmooth function $\ell \circ h$?

Unfortunately, the answer to these questions is "no", as we show below. First, we derive the lower iteration bound for any algorithm with access only to a subgradient oracle in the interpolation setting.

---

**Theorem 4.6** (Lower bound with interpolation). *Given $t < d$ and positive constants $L$ and $R$, and an initial vector $w^{(1)}$. Let $\ell = \frac{1}{2}(\cdot)^2$. Then there exists an L-Lipschitz function $h$ such that $f := l \circ h$ is convex, $f^* = \min_w f(w) = 0$, $\|w^{(1)} - w^*\| \leq R$ such that*

$$\min_{1 \leq s \leq t} f(w^{(s)}) - f^* \geq \frac{L^2 R^2}{2(t+1)}. \tag{4.15}$$

---

*Proof.* With out loss of generality, we assume the initial point $w^{(1)} = 0$. Let $h(w) = L\|w - w^*\|_\infty$, and

$$w^* = \Big[ \underbrace{\frac{R}{\sqrt{t+1}} + \epsilon, \frac{R}{\sqrt{t+1}} + \frac{\epsilon}{2}, \ldots, \frac{R}{\sqrt{t+1}} + \frac{\epsilon}{2^t}}_{t+1 \text{ entries}}, \quad \underbrace{0, \ldots, 0}_{(d-t-1) \text{ entries}} \Big],$$

where $\epsilon$ is some arbitrary constant greater than 0. It is easy to check that $\ell$ is 1-smooth, $h$ is $L$-Lipschitz, $f = \ell \circ h$ is convex, and $f^* = 0$.

We follow Nemirovski & Yudin (1983) and assume that any algorithm that uses only subgradient information generates iterates $w^{(s)} \in \text{span}\{\partial f(w^{(1)}), \ldots, \partial f(w^{(t)})\}$ for all $s \leq t$. By our construction, we can further obtain $w^{(s)} \in \text{span}\{e_1, e_2, \ldots, e_t\}$ for all $s \leq t$. Therefore, for all $s \leq t$,

$$f(w^{(s)}) - f^* \geq \frac{1}{2} \left( \frac{LR}{\sqrt{t+1}} + \frac{L\epsilon}{2^{s-1}} \right)^2.$$

Note that the above holds $\forall \epsilon > 0$. Taking $\epsilon \to 0^+$, we completes the proof. □

Theorem 4.6 established $\Omega(1/\epsilon)$ iteration complexity for subgradient method under interpolation. Combining this result with the $\mathcal{O}(1/\epsilon)$ iteration complexity in Corollary 4.4, we can conclude that the rate $\mathcal{O}(1/\epsilon)$ is optimal in the interpolation setting and no acceleration is possible for subgradient-based methods. Then we proceed to the lower bound with assuming interpolation.

---

**Theorem 4.7** (Lower bound without interpolation). *Given $t < d$, $L, R > 0$ and an initial point $w^{(1)}$. Let $\ell(\cdot) = \frac{1}{2}(\cdot)^2$, and there exist $L$-Lipschitz function $h$ satisfying $f := l \circ h$ is convex, $\|w^{(1)} - w^*\| \leq R$ such that*

$$\min_{1 \leq s \leq t} f(w^{(s)}) - f^* \geq \frac{L^2 R^2}{\sqrt{t+1}}. \tag{4.16}$$

---

*Proof.* Similar to the proof of Theorem 4.6, we set $w^{(1)} = 0$. But we change the construction of $h(w)$ to $h(w) = L(\|w - w^*\|_\infty + R)$, and $w^*$ is defined in the same way as in the proof of Theorem 4.6:

$$w^* = \Big[ \underbrace{\frac{R}{\sqrt{t+1}} + \epsilon, \frac{R}{\sqrt{t+1}} + \frac{\epsilon}{2}, \ldots, \frac{R}{\sqrt{t+1}} + \frac{\epsilon}{2^t}}_{t+1 \text{ entries}}, \underbrace{0, \ldots, 0}_{(d-t-1) \text{ entries}} \Big],$$

where $\epsilon$ is some arbitrary constant greater than 0. Following the same analysis of the proof of Theorem 4.6. $w^{(s)} \in \text{span}\{e_1, e_2, \ldots, e_t\}$ $\forall s \leq t$ and

$$f(w^{(s)}) - f^* \geq \frac{L^2}{2} \left( \frac{R}{\sqrt{t+1}} + \frac{\epsilon}{2^{s-1}} + R \right)^2 - \frac{L^2 R^2}{2} \geq \frac{L^2 R^2}{\sqrt{t+1}} \qquad \forall s \leq t,$$

By taking $\epsilon \to 0^+$, we have $\|w^{(1)} - w^*\| \leq R$ and completes the proof. □

Theorem 4.7 provides an $\Omega(1/\epsilon^2)$ iteration complexity for subgradient-based methods for solving the structured function $\ell \circ h$. This matches the lower bound for solving general nonsmooth objective function with subgradient-based method, and implies that the structure $\ell \circ h$ itself without interpolation cannot give us an improved iteration complexity of subgradient-based methods.

## 5 NUMERICAL EXPERIMENTS

The smoothness of the loss function $\ell$ is crucial to our analysis. We now present some numerical experiments to compare the convergence of SSGD for training ReLU neural networks with smooth and nonsmooth loss functions. We denote $\{x_i\}_{i=1}^n$ as training samples and $\{y_i\}_{i=1}^n$ as training labels. $\hat{y}_i$ stands for the prediction of the $i$-th sample $x_i$ from the trained model.

### 5.1 TEACHER-STUDENT SETUP

We randomly generate a small one hidden layer neural network with 16 neurons and ReLU activation as the teacher network, the network takes 16 dimensional vectors as inputs and outputs a scalar. Then we generate 128 random vectors from the Gaussian distribution as our training data $\{x_i\}_{i=1}^{128} \subset \mathbb{R}^{16}$ and get their corresponding labels $\{y_i\}_{i=1}^{128}$ as the output of the teacher network. In order to ease the training and satisfy the interpolation assumption, we overparameterize the student neural network

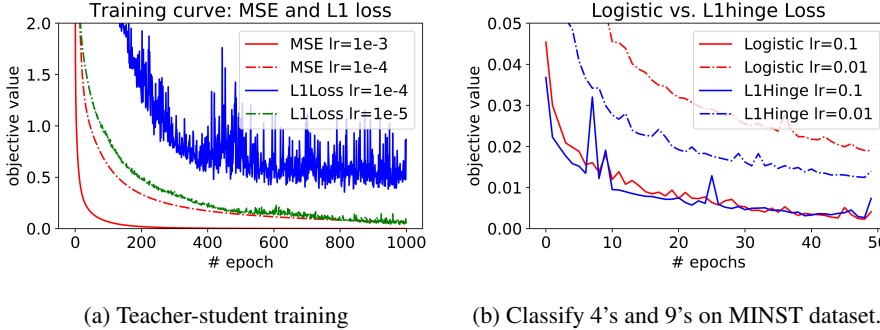

(a) Teacher-student training  (b) Classify 4's and 9's on MINST dataset.

Figure 5.1: The performance of SSGD with smooth and nonsmooth loss functions.

and set it to be a one hidden layer network with 512 neurons and ReLU activation. We train the student network with different loss functions: squared loss e.g., $\frac{1}{n}\sum_{i=1}^n(y_i-\hat{y}_i)^2$ and absolute loss e.g., $\frac{1}{n}\sum_{i=1}^n|y_i-\hat{y}_i|$ with different learning rates. The training curves are shown in Figure 5.1a. We can observe that the training curve with squared loss is smoother than the curve with absolute loss. For absolute loss, the performance of SSGD is more sensitive to the change of learning rate and we need to decrease the learning rate to obtain a lower objective value. These observations validate the importance of the smoothness of loss function under the interpolation setting.

## 5.2 CLASSIFY 4'S AND 9'S ON MNIST DATASET

We train the LeNet (Lecun et al., 1998) on the MNIST dataset to classify 4's and 9's. To convert this task to a binary classification problem, we transform the labels $\{y_i\}_{i=1}^n$ to $\{-1,+1\}^n$. Then we run SSGD to train the model with difference loss functions: logistic loss e.g., $\frac{1}{n}\sum_{i=1}^n\log(1+\exp(-y_i\hat{y}_i))$ and L1-hinge loss e.g., $\frac{1}{n}\sum_{i=1}^n\max\{0,1-y_i\hat{y}_i\}$ and with different learning rates. The training curves are presented in Figure 5.1b. Different from the teacher-student experiment, SSGD in this task perform similarly with smooth and nonsmooth loss functions. We conjecture that this is because the objective function with L1-hinge loss almost satisfies Proposition 4.1 locally at the solution, namely eq. (4.1) holds for most training samples in a neighbourhood of the solution. Our observation supports this conjecture. We observe that more that 95% of our final predictions $\hat{y}_i$'s satisfy the condition $|\hat{y}_i|>2$. Since the L1-hinge loss is locally smooth when $|\hat{y}_i|>2$, we can thus say that most training samples satisfy eq. (4.1) locally at the solution. While for the teacher-student training problem, its objective is nonsmooth at solution (when zero residual is attained) since the absolute value function $\ell(x):=|x|$ is nonsmooth at 0. Therefore the objective of the teacher-student training problem does not satisfy Proposition 4.1 locally as solution and running SSGD to solve it could suffer from slow convergence.

## 6 CONCLUSION AND DISCUSSION

An empirical minimization problem based on composite functions has sufficient structure to allow for a tight convergence analysis that explains the effectiveness of stochastic subgradient descent methods on nonsmooth problems with interpolation. Surprisingly, the complexity bounds $\mathcal{O}(1/\epsilon)$ and $\mathcal{O}(\log(1/\epsilon))$ that we prove under interpolation match those of stochastic gradient descent for smooth functions.

Our convergence analysis is based on the convexity properties of $f$. As we mention in connection with Theorem 4.5, the strong convexity assumption can be relaxed to weaker conditions, but even these exclude important nonconvex models that appear in neural networks. It is still an open problem as to whether a linear convergence rate for subgradient methods under a weaker assumptions, such as the restricted secant inequality (RSI). In Section 4.3 we proved that the rate $\mathcal{O}(1/\epsilon)$ is optimal for subgradient-based methods under interpolation. However, this lower bound holds only for subgradient-based algorithms. Smoothing techniques based on Moreau envelopes (Nesterov, 2005) can sometimes lead to acceleration for nonsmooth optimization, which may be further avenue to explore for obtaining an accelerated SSGD method.

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

## APPENDIX

## A  THE PROOF OF PROPOSITION 4.2

*Proof of Proposition 4.2.* Given $w_1, w_2 \in \mathbb{R}^d$, $\forall i \in [n]$,

$$\|\partial f_i(w_2) - \partial f_i(w_1)\|$$
$$\leq \|\ell'(h_i(w_2))\partial h_i(w_2) - \ell'(h_i(w_1))\partial h_i(w_1)\|$$
$$\leq \|\ell'(h_i(w_2))\partial h_i(w_2) - \ell'(h_i(w_2))\partial h_i(w_1) + \ell'(h_i(w_2))\partial h_i(w_1) - \ell'(h_i(w_1))\partial h_i(w_1)\|$$
$$\leq \|\ell'(h_i(w_2))\partial h_i(w_2) - \ell'(h_i(w_2))\partial h_i(w_1)\| + \|\ell'(h_i(w_2))\partial h_i(w_1) - \ell'(h_i(w_1))\partial h_i(w_1)\|$$
$$\leq \|\ell'(h_i(w_2))(\partial h_i(w_2) - \partial h_i(w_1))\| + \|(\ell'(h_i(w_2)) - \ell'(h_i(w_1)))\partial h_i(w_1)\|$$
$$\leq |\ell'(h_i(w_2))|\|\partial h_i(w_2) - \partial h_i(w_1)\| + |\ell'(h_i(w_2)) - \ell'(h_i(w_1))|\|\partial h_i(w_1)\|$$
$$\leq 2L|\ell'(h_i(w_2))| + |h_i(w_2) - h_i(w_1)| \times L$$
$$\overset{(i)}{\leq} 2L\sqrt{2f_i(w_2)} + L^2\|w_2 - w_1\|, \tag{A.1}$$

where $(i)$ follows the same argument as the proof of Proposition 4.1:

$$|\ell'(h_i(w_2))| = |\ell'(h_i(w_2)) - \ell'(0)| \leq \sqrt{2(\ell(h_i(w_2)) - \ell(0))} = \sqrt{2f_i(w_2)}.$$

Exchange $w_1$ and $w_2$, we can also get

$$\|\partial f_i(w_2) - \partial f_i(w_1)\| \leq 2L\sqrt{2f_i(w_1)} + L^2\|w_2 - w_1\|. \tag{A.2}$$

Combining Equation (A.2) and Equation (A.1), we finished the proof for Equation (4.4).

Given $w_1, w_2 \in \mathbb{R}^d$, note that $f_i(w)$ is almost every differentiable by Rademacher's Theorem, thus

$$f_i(w_2) = f_i(w_1) + \int_0^1 \langle \partial f_i(w_1 + \tau(w_2 - w_1)), w_2 - w_1 \rangle d\tau$$

$$= f_i(w_1) + \langle \partial f_i(w_1), w_2 - w_1 \rangle + \int_0^1 \langle \partial f_i(w_1 + \tau(w_2 - w_1)) - \partial f_i(w_1), w_2 - w_1 \rangle d\tau \tag{A.3}$$

Note that

$$\int_0^1 \langle \partial f_i(w_1 + \tau(w_2 - w_1)) - \partial f_i(w_1), w_2 - w_1 \rangle d\tau$$

$$\leq \int_0^1 \|\partial f_i(w_1 + \tau(w_2 - w_1)) - \partial f_i(w_1)\| \|w_2 - w_1\| d\tau$$

$$\leq \int_0^1 \left(2L\sqrt{2f_i(w_1)} + L^2\|\tau(w_2 - w_1)\|\right) \|w_2 - w_1\| d\tau$$

$$\leq 2L\|w_2 - w_1\|\sqrt{2f_i(w_1)} + \frac{L^2}{2}\|w_2 - w_1\|^2. \tag{A.4}$$

Plug Equation (A.4) into Equation (A.3), we finished the proof for Equation (4.5). $\qquad\square$

## B    ANOTHER CHOICE OF LEARNING RATE SCHEDULING

The following theorem is based on the result of Stich (2019)'s analysis on refined learning rate scheduling. Note that the refined learning rate scheduling depends not only on the Lipschitz constant $L$, but also rely on the number of iteration $T$ (we need the number of iterations as an input of our algorithm).

**Theorem B.1.** *Assume that $f$ is $\mu$-strongly convex for some $\mu \geq 0$. For all positive integer $T$, there exist a constant learning rate scheduling $\alpha_t := \alpha \leq 1/(2L^2)$, such that the SSGD iterates $w^{(t)}$ satisfy*

$$\min_{t \in [T+1]} \mathbb{E}[f(\bar{w}^{(t)}) - f^*] + \mu\mathbb{E}[\|w^{(T+2)} - w^*\|^2] \leq 64L^2\|w^{(1)} - w^*\|^2 \exp\left(-\frac{\mu T}{4L^2}\right) + \frac{72L^2 f^*}{\mu T} \tag{B.1}$$

*for $\mu > 0$, and*

$$\min_{t \in [T]} \mathbb{E}[f(\bar{w}^{(t)}) - f^*] \leq \frac{4L^2\|w^{(1)} - w^*\|^2}{T} + \frac{4L^2 f^*\|w^{(1)} - w^*\|}{\sqrt{T}} \tag{B.2}$$

*for $\mu = 0$.*

*Proof.* We start with eq. (4.12),

$$eq. (4.12)$$

$$= (1 - \mu\alpha_t)\|w^{(t)} - w^*\|^2 - 2\alpha_t(f(w^{(t)}) - f^*) + 2\alpha_t^2 L^2 f(w^{(t)})$$

$$= (1 - \mu\alpha_t)\|w^{(t)} - w^*\|^2 - 2(\alpha_t - \alpha_t^2 L^2)(f(w^{(t)}) - f^*) + 2\alpha_t^2 L^2 f^*$$

$$\overset{(i)}{\leq} (1 - \mu\alpha_t)\|w^{(t)} - w^*\|^2 - \alpha_t(f(w^{(t)}) - f^*) + 2\alpha_t^2 L^2 f^*, \tag{B.3}$$

where (i) is from the fact that $2(\alpha_t - \alpha_t^2 L^2) \geq \alpha_t$ given that $0 < \alpha_t \leq 1/(2L^2)$. Note that eq. (B.3) is in the same form as Eq. (8) from Stich (2019)'s work, then we can directly apply (Stich, 2019, Lemma 2, Lemma 3): there exist a learning rate scheduling such that

$$\min_{t \in [T+1]} \mathbb{E}[f(\bar{w}^{(t)}) - f^*] + \mu\mathbb{E}[\|w^{(T+2)} - w^*\|^2] \leq 64L^2\|w^{(1)} - w^*\|^2 \exp\left(-\frac{\mu T}{4L^2}\right) + \frac{72L^2 f^*}{\mu T}. \tag{B.4}$$

Then we can obtain eq. (B.2) by applying (Stich & Praneeth Karimireddy, 2019, Lemma 13) and therefore completes the proof. Note that our proof is mostly based on the recurrence relation eq. (B.3) and Stich (2019)'s analysis, we refer interested readers to Stich (2019)'s work for more technical details.

$\qquad\square$

