# OpenReview forum: "Fast convergence of stochastic subgradient method under interpolation"
_ICLR.cc/2021/Conference — ICLR 2021 Poster_

### Official Review · AnonReviewer1 · 2020-10-27
**A borderline paper, closer to acceptance**

**Rating:** 7
**Confidence:** 4

**Review:**

Summary:
The paper considers stochastic finite-sum minimization with a special structure of composition of a smooth univariate loss with a non-smooth Lipschitz function. A generalized gradient bound is proved for this setting as well semi-smoothness. Using the former bound, sublinear 1/k rate is proved for stochastic subgradient method under convexity and interpolation conditions. With additional assumption of strong convexity, a linear convergence rate is proved.  Further, a lower complexity bound is claimed for the considered class of functions, which proves that the obtained bounds for the SSGD are optimal.

Evaluation:
In my opinion, the paper is a borderline, slightly on the acceptance side. The obtained convergence rates are interesting. On the other hand, the convex setting is not well motivated by applications, and the lower bounds are obtained under quite restrictive assumption on the possible algorithms.

Pros:
1. Faster convergence rates for SSGD under interpolation condition, which match the non-accelerated bounds for the smooth case.
2. This can shed some light on why SSGD works quite fast in practice.
3. Proposition 4.1 shows that this setting is very close to the smooth optimization case.

Cons:
1. It is not clear what could be particular real examples of problems in the convex case which are covered by the considered framework.
2. Since there is almost sure differentiability, it may happen that SSGD visits only points of smoothness and, thus, generates the same trajectory as SGD. This could be a simpler explanation of why SSGD works very similarly to SGD in practice.
3. The lower bounds are proved under a restrictive assumption that $w^{(s)} \in span \{e_1, e_2,...,e_t\}$. This is not what assumed in Nemirovski & Yudin (1983). Instead, standard way is to assume that $w^{(s)}$ belongs to the span of previous subgradients. Then, with a special construction of resisting oracle, it is proved that only one non-zero component can be added per iteration.
4. No numerical experiments to support the theoretical findings.

Minor comments:
1. Some related literature is not mentioned. There is a set of works on so-called compositional optimization. See https://link.springer.com/article/10.1007/s10107-018-1311-3 , references therein and also papers which cite this paper.
2. There are some misprints.
a. Last but one line of the first paragraph of Sect. 2.1. "complexity of get" -> "complexity of getting an"
b. Last but one line of the first paragraph of Sect. 3.  "2-norm a vector" -> "2-norm of a vector"
c. First line of the second paragraph of Sect. 5. extra "the".

---

> ### Author Response · Authors · 2020-11-14
> **Thanks for your useful feedback**
>
> Thanks for your helpful comments.
>
> **Q: A particular example of a convex objective function.**
>
> A: We agree that convexity is a strong assumption that does not hold for neural networks in general. Recent progress in wide neural networks indicates that neural networks can have some local convexity property with random initialization and massive overparameterization (the width the neural network is set to be very large) [1,2,3], see also [Lemma 4.2, 4]. In order to obtain iteration complexity to approach global minimizer, convexity, or local convexity is usually needed. When convexity or local convexity is not allowed, most convergence analysis in the literature can only guarantee iteration complexity to approach a stationary point instead of a global minimizer, but one will need to assume smoothness in this setting and therefore does not apply to our work. The local convexity property of the objective function usually varies from task to task. Our purpose here is to develop an analysis for general unsmooth minimization that satisfies the composition structure, so we assume convexity for simplicity. But again we agree that convexity is a strong assumption that may not hold for a wide range of problems.
>
> [1] Arthur Jacot, Cl{\'{e}}ment Hongler, Franck Gabriel. Neural Tangent Kernel: Convergence and Generalization in Neural Networks, NeurIPS 2018.
>
> [2] Zeyuan Allen{-}Zhu, Yuanzhi Li, Zhao Song. A Convergence Theory for Deep Learning via Over-Parameterization. ICML 2019.
>
> [3] Simon S. Du, Jason D. Lee, Haochuan Li, Liwei Wang, Xiyu Zhai. Gradient Descent Finds Global Minima of Deep Neural Networks. ICML 2019.
>
> [4] Yuan Cao, Quanquan Gu. Generalization Bounds of Stochastic Gradient Descent for Wide and Deep Neural Networks. NeurIPS 2019.
>
> **Q: Another intuitive explanation of the effectiveness of SSGD.**
>
> A: We agree that the objective function is almost everywhere differentiable assuming L-Lipschitz continuous. But the iteration complexity of running gradient descent on an everywhere differentiable function is different from the iteration complexity for a smooth function. One example is the absolute value function $f(x) = |x|$ (which is almost everywhere differentiable), we need $ O( 1 / \epsilon^2 ) $ iterations to get an $\epsilon$-approximate solution when running subgradient descent. However, for smooth function such as the logistic loss, we only need $O(1/\epsilon)$ gradient descent steps to obtain an $\epsilon$-approximate solution.
>
>
> **Q: Proof for the lower bound.**
>
> A: Thanks for pointing out the slight flaw in the proof of Theorem 4.6. We fixed this issue using the technique given by AnonReviewer3.
>
> Thanks for pointing out some typos in our manuscript, we have fixed them in the updated version. Also, we appreciate you for mentioning another line of works on compositional optimization, we have included some works in this line into our discussion in the related work section.
>
> **Q: Empirical experiments?**
>
> A: Thanks for pointing out the lack of empirical evaluation, we included some numerical experiments in the updated manuscript.

---

> > ### Comment · AnonReviewer1 · 2020-11-23
> > **Thanks for the answers, but they do not seem completely convincing to me**
> >
> > I would like to thank the authors for their answers and for the update of their manuscript, especially for adding a numerical illustration.
> >
> > Yet the answers to the following questions do not seem completely convincing to me.
> >
> > Q: Another intuitive explanation of the effectiveness of SSGD. The arguments provided in the answer refer to the theory, whereas my comment was about performance in practice.
> >
> > Q: Proof for the lower bound. The lower bounds are still proved  in the updated version of the paper under a restrictive assumption that $w^{(s)} \in {\rm span} \{e_1,...,e_t\}$. I believe that this limits the generality of the result.
> > Moreover, in the constructed example in Theorem 4.6, $\ell$ is 2-smooth rather than $1$-smooth as it is written.
> > Also, why does the equality for $f(w^{(s)})-f^*$ in the end of the proof hold?
> > It should be underlined that $h$ is $L$-Lipschitz w.r.t. infinity norm, rather than $2$-norm.
> >
> > Thus, I would prefer to keep my score unchanged.

---

> > > ### Author Response · Authors · 2020-11-24
> > > **Thanks for your feedback**
> > >
> > > Thanks for your clarification on your comments. We updated the proof of Theorem 4.6 and 4.7 in the manuscript according to your comments and here are some answers regarding your questions:
> > > * Yes, the $\ell$ in the old version is 2-smooth. We updated $\ell$ to $\frac{1}{2} (\cdot)^2$ and changed some other constants by the factor $\frac{1}{2}$ in the new version. We appreciate your careful proofreading.
> > > * Yes, the black-box model assumes $w^{(s)} \in span~ \{ \partial f( w^{(1)} ), \ldots, \partial f( w^{(t)} ) \}$, and $ w^{(s)} \in span~ \{
> > > e_1, e_2, \ldots, e_t \} $ is from the construction of the problem. We updated our proof to address this issue.
> > > * We do mean $h$ to be $L$-Lipschitz continuous w.r.t. 2-norm. By norm inequality, we know that $|| x ||_\infty \leq || x ||_2$. So if a function is $L$-Lipschitz w.r.t. infinity norm, it must also be $L$-Lipschitz w.r.t. 2-norm.
> > > * $f(w^{(s)}) - f^*$ should be followed by inequality instead of equality. Thanks again for your careful proof-reading.

---

> > > > ### Comment · AnonReviewer1 · 2020-11-25
> > > > **Thanks for the update. Now I can increase the score.**
> > > >
> > > > I would like to thank the authors for the clarification. My concerns are addressed and I can increase the score.

---

### Official Review · AnonReviewer4 · 2020-10-28
**Convergence analysis of stochastic subgradient descent**

**Rating:** 6
**Confidence:** 4

**Review:**

Summary:

This paper considers the behavior of the stochastic subgradient descent method under the interpolation condition. They provide convergence rates for both convex and strongly-convex objectives.

Reasons for score:

This paper is well-written. My major concern is the novelty. In terms of both methodology and theory, the paper seems to be an extension of classical results. While the theoretical results are interesting and useful, they are fairly straightforward. The authors should also provide comparisons between their method and previous approaches both theoretically and numerically. Some detailed comments are listed below. Hopefully, the authors can address my concern in the rebuttal period.

Here are my main comments:

1) The authors need to claim clearly what are their theoretical contributions. The main results Theorem 4.3 and Theorem 4.5 seem to be easy extensions of traditional SGD convergence analysis.

2) Interpolation condition is a core concept in this paper. However, there is no clear statement of it. It only appears in some comments after theorems. I suggest the authors provide a formal condition at the beginning of their main results and offer some intuition behind it.

3) The assumption that f is convex is not very intuitive. Could the author give some examples to illustrate that? Moreover, the term "overparameterized" has been brought up several times in the abstract and the introduction part. It would be better to give an example of an overparameterized model, which also helps to explain the convexity assumption.

4) Some numerical experiments would be preferred to better illustrate the theoretical results.

---

> ### Author Response · Authors · 2020-11-14
> **Thanks for your useful feedback**
>
> Thanks for your helpful comments.
>
> **Q: What are the contributions of this paper?**
>
> A: Our contributions:
> * The best known iteration complexity for SSGD is $O(1/\epsilon^2)$ and $ \tilde{O} ( 1/\epsilon ) $ ($\tilde{O}(\cdot)$ omits the polylog term) for convex and strongly convex objectives when interpolation holds. In this work, by assuming a composite structure of the objective function with a smooth outer function, we improve these rates to $O( 1/\epsilon )$ and $ O(\log( 1/\epsilon )) $ in the convex and strongly convex setting under interpolation. These rates match the rates of SGD for smooth objectives and therefore partially explains why training machine learning models with nonsmooth objective can sometimes be as fast as training smooth objectives in practice.
> * Our Proposition 4.1 and Proposition 4.2 can be beneficial to the optimization community. These two propositions hold with mild assumptions: the composite structure; the loss function $\ell$ is "nice" (1-dimensional convex, smooth, nonnegative and infimum is 0); the inner functions $h_i$'s are Lipschitz continuous. We don't need to assume the objective function to be convex or smooth, which is more general than the current result in the literature [Proposition 2, 1]. Thanks for pointing out the ambiguity of our contributions, we updated a more detailed description of our contribution in the introduction section in the updated manuscript.
> * We totally agree that the proof of our Theorem 4.3 and Theorem 4.5 can be viewed as an extension of the traditional subgradient descent analysis because our proof is indeed based on some careful modifications of the classic result. Although our proof techniques are similar to the classic result, the conclusion is very different. Comparing with the classic result [Equation 2, 2], we are able the replace their $G^2$ (their Lipschitz constant) with $f^*$ (as shown in our equation 4.6) by using our Proposition 4.1. Therefore we are able to improve the classic convergence rate $O(1/\epsilon^2)$ to $O(1/\epsilon)$ under interpolation.
>
> [1] Sharan Vaswani, Francis Bach, Mark Schmidt. Fast and Faster Convergence of {SGD} for Over-Parameterized Models and an Accelerated Perceptron. Proceedings of AISTATS, 2019.
>
> [2] Stephen Boyd, Lin Xiao, Almir Mutapcic. Subgradient methods. Lecture notes of EE392o, Standford University.
>
>
>
> **Q: The concept of interpolation.**
>
> A: Thanks for pointing out the confusion on the concept of interpolation. Some of our results do not need to assume interpolation (Proposition 4.1, Proposition 4.2, and Theorem 4.3). The interpolation assumption appears in our Corollary 4.4. We updated a discussion on interpolation in the problem setting section. Thanks again for your suggestion to make this clarification.
>
>
> **Q: Assumption of convexity and the example of the overparameterized model.**
>
> A: We agree that assuming convexity is a strong assumption. One example of local convexity is the over-parameterized neural networks (the width of the neural network is set to be very large). AnonReviewer1 also raised a similar question, please see our response to his/her question for a more detailed explanation. Thanks again for your helpful comment and let us know if you need more clarification on this question.
>
> **Q: Empirical experiments?**
>
> A: Thanks for pointing out the lack of empirical evaluation, we included some numerical experiments in the updated manuscript. Please let us know if you have any new comments regarding the numerical experiments section.

---

> > ### Comment · AnonReviewer4 · 2020-11-20
> > **Thank you for the response and modification**
> >
> > Thank you for your response and modification. The theoretical contribution is clearly stated right now and empirical results have been added to illustrate your theoretical findings. I have raised my rating accordingly.
> >
> > However, I still think the theoretical contribution is limited and the authors' response to the convexity assumption does not fully convince me.
> >
> > My concern about the convex and strongly-convex assumption is not about whether convexity is too strong in real-world applications. Instead, I would like to know if you can give an example, that $h(\cdot)$ is non-smooth and Lipschitz, $\ell(\cdot)$ is convex and 1-smooth and $f = \ell(h(\cdot))$ is a convex function.
> >
> > I will raise my rating again if you can give a truly convex example (even in some trivial learning tasks).

---

> > > ### Author Response · Authors · 2020-11-20
> > > **Thanks for your response**
> > >
> > > Thanks for your response. We do can construct a simple 1-dimensional example to satisfy our assumption. Consider $ \ell( \cdot ) = ( \cdot )^2 $ and $ h(w) = -w $ when $ w \leq 1 $ and $h(w) = -2w$ when $w > 1$. It is easy to check that $h(w)$ is nonconvex and nonsmooth. In this case $f(w) = \ell( h(w) ) = w^2$ when $w \leq 1$ and $f(w) = 4w^2$ when $w > 1$ is convex (but not smooth at $w=1$).
> > >
> > > A slightly more interesting example is to set $h(w) = -w$ when $w \leq \delta$ and $h(w) = -2w$ when $w > \delta$ for some $\delta > 0$. Then $f(w) = w^2$ when $w\leq \delta$ and $f(w)=4w^2$ when $w > \delta$. By letting $\delta \to 0$, the function $f$ is nonsmooth near its solution $w^*=0$ but we can still get the convergence rate $O( 1/\epsilon )$.
> > >
> > > Thanks for raising this question, we will consider adding this simple example to the Appendix. Please let us know if you have any other questions.

---

### Official Review · AnonReviewer3 · 2020-10-28
**New complexity bounds for stochastic non-smooth (with a certain structure) optimization under interpolation**

**Rating:** 8
**Confidence:** 5

**Review:**

## Summary
The paper focuses on the empirical risk minimization problem when summands have a form of composition of smooth non-negative convex and non-smooth functions under interpolation condition, i.e. when the loss is zero in the solution. In this setting, authors derived convergence rates $O\left(\frac{1}{\varepsilon}\right)$ and $O\left(\log\left(\frac{1}{\varepsilon}\right)\right)$ for Stochastic Subgradient Descent (SSGD) for convex and strongly convex objectives respectively. Up to the difference in factors hidden in $O(\cdot)$, these rates match the rate of SGD in the convex and strongly convex cases under interpolation assumption, which partially explains why neural networks with non-smooth activations can usually be trained as fast as ones with smooth activations. Moreover, the authors derived lower complexity bounds for this particular class of problems showing that SSGD is optimal in the convex case under interpolation.

Overall, the contributions are quite impressive and solid, the paper is clearly written, and the proofs are simple and short (which is actually one of the biggest advantages of this paper). However, there are also some technical issues in the proofs of lower bounds. Fortunately, these issues can be easily fixed (see the Weaknesses part for further details).

## Strengths
1. **Clarity, motivation, and related work.** The paper is reader-friendly: the results are presented clearly and explained well, the problem is well-motivated, the literature review is very concrete and covers the most relevant papers, to the best of my knowledge.
2. **Lower bounds and tight rates for SSGD.** Authors proposed lower bounds showing that $O(1/\varepsilon)$ is the best convergence rate one can hope for in this setting. These lower bounds are tighter than standard lower bounds for non-smooth convex optimization, and they rely on a particular structure of function $f$. Moreover, the authors show that SSGD complexity bounds match the proposed lower bounds.
3. **Simple proofs.** I have checked all proofs: they are mathematically correct **(except the proofs of lower bounds; see the weaknesses part)**, clearly written, and short. Taking into account the novelty and importance of the proposed results, simple and solid proofs are a nice bonus making the paper easy to read.

## Weaknesses
1. **Experiments.** There are almost no experiments in the paper. While the contribution of the paper is primarily theoretical, I believe the paper would benefit from adding some numerical study justifying the proposed theory on different problems.
2. **Inaccuracies in the proofs of lower bounds.** Formally speaking, the proof of Theorem 4.6 is incorrect. Indeed, if $w^{(0)} = 0$, then $w^{(1)} \not\in \text{span}(e_1,\ldots, e_t)$, but $w^{(1)} \in \text{span}(e_1,\ldots, e_t, {\color{red}e_{t+1}})$, because the subgradient of $\partial f(w^{(0)}) = \frac{2L^2R^2}{t+1}\cdot\text{conv}(e_1,\ldots,  e_t, {\color{red}e_{t+1}})$ . Since the proof of Theorem 4.7 relies on the following technique, it is mathematically incorrect too. However, it can be easily fixed without changing the main idea; see my version of the proof below.
 - **Proof of Theorem 4.6.** Let us consider the same function $f$, the same starting point $w^{(0)} = 0$, but change the definition of $w^{\star}$ in the following way: $w^{\star}=\left[\frac{R}{\sqrt{t+1}} + \varepsilon, \frac{R}{\sqrt{t+1}} + \frac{\varepsilon}{2}, \ldots, \frac{R}{\sqrt{t+1}} + \frac{\varepsilon}{2^t}, 0, \ldots, 0\right]^\top$, where $\varepsilon > 0$ is some small number. We have $\partial f(w) = 2L^2||w - w^{\star}||_{\infty}\cdot \text{conv}( \partial(|w_i - {w_i}^{\star}|) \quad|\quad  i: \quad i\in I(w) )$, where $i\in I(w)$ means that  $|w_i-{w_i}^{\star}|$ equals $\ell_\infty$-norm of $w-w^*$. Therefore, one can easily check via induction that $w^{(s)}$ lies in the span of first $s$ coordinate vectors for all $s \leq t$ since coordinates $w_i^*$ of $w^*$ are decreasing when $i$ increases (see also the formula for $\partial f(w)$). Using this we derive $f(w^{(s)}) - f(w^*) \geq \left(\frac{LR}{\sqrt{t+1}} + \frac{L\varepsilon}{2^s}\right)^2$ for all $s \leq t$. Taking the limit w.r.t. $\varepsilon \downarrow 0$ we get the result.

## Questions and Comments
1. page 1, Abstract, "By leveraging ...": The sentence is incomplete and should be rewritten.
2. page 4, "$f$ is differentiable ...": I guess, this sentence is slightly misleading: $f(w) = 0$ means that $w$ is a minimizer and interpolation condition is satisfied, while in general if $w$ is a minimizer, then it is not necessary that $f(w) = 0$ (unless we have an interpolation). Consider rewriting of the sentence.
3. page 11, inequality between (A.1) and (A.2): $\sqrt{2\left(l(h_i(w_2) - l(0)\right)}$ $\to$ $\sqrt{2\left(l(h_i(w_2){\color{red})} - l(0)\right)}$
4. page 11, ineq. (A.3): $\langle \partial f_i(w_1 + \tau(w_2-w_1)) - \partial f(w_1), w_2 - w_1 \rangle$ $\to$ $\langle \partial f_i(w_1 + \tau(w_2-w_1)) - \partial f_{\color{red}i}(w_1), w_2 - w_1 \rangle$
5. page 5, proof of Theorem 4.3: The notation $g_i^{(t)}$ is not introduced, while can be recovered from the context. Anyway, authors should add a formal definition.
6. page 6, Theorem 4.5: Actually, one can extend the result using the recurrence (4.12) for the general choice of $\alpha_t$ and applying some standard stepsize schedules for SGD; e.g., see
 - Stich, Sebastian U. "Unified optimal analysis of the (stochastic) gradient method." arXiv preprint arXiv:1907.04232 (2019).

## Final Remarks
To conclude, the paper makes an important contribution to the theory of stochastic optimization under interpolation. There are small issues in the proofs for lower bounds that can be easily fixed (see my comment in the Weaknesses part). I encourage the authors to apply the needed corrections. **If the authors fix the issues mentioned above, I will increase my score.**

---

> ### Author Response · Authors · 2020-11-14
> **Thanks for your useful feedback**
>
> Thanks for your helpful comments. We appreciate that you point out the slight flaw in our proof and provide a fix. Your fix is totally valid and we have updated our manuscript accordingly.
>
> Thanks for pointing out some typos and confusing sentences. We have fixed them in our updated manuscript.
>
> Thanks for mentioning other choices of stepsize scheduling, I think we cannot directly apply Stich's [1] technique because our Eq (4.12) does not satisfy his Eq (8). When constant learning rate is used, $\frac{4}{\mu} f^*$ will always appear in Eq (4.13) (this term does not depend on the choice of stepsize), it seems nontrivial to obtain a rate similar to the one in [Theorem 5, 1]. Please correct us if we misunderstood your comments. Thanks again for your insightful suggestion.
>
> [1] Sebastian U. Stich. Unified optimal analysis of the (stochastic) gradient method. arXiv preprint 2019.
>
> Some numerical experiments are also included in the updated manuscript. Please let us know if you have any new comments on the numerical experiments.

---

> > ### Comment · AnonReviewer3 · 2020-11-14
> > **Thanks for the response & clarification about stepsize scheduling**
> >
> > **First of all, I want to thank the authors for the updates** they applied according to my suggestions and also for adding experimental results. Now the paper looks much better and, in particular, does not contain mathematical inaccuracies. I am increasing my score from 6 to 8.
> >
> > **Next, I want to clarify my comment about stepsize scheduling.** In fact, one can stop the derivation in (4.12) on step (iii) and instead of choosing $\alpha_t = \frac{1}{L^2}$ one can keep it there and say that $\alpha_t \leq \frac{1}{L^2}$ and get in the RHS $2 \alpha_{t}^{2} L^{2} f\left(w^{*}\right)$. For the obtained recurrence, one can apply stepsize scheduling similar to ones discussed in [1]. Using this, one can get provable convergence to any accuracy even when the model is not overparameterized.
> >
> > [1] Sebastian U. Stich. Unified optimal analysis of the (stochastic) gradient method. arXiv preprint, 2019.

---

> > > ### Author Response · Authors · 2020-11-16
> > > **Thanks for your clarification**
> > >
> > > Thanks for your clarification. Due to space limitation, we added some discussion on this more refined learning rate scheduling in the Appendix, the result is mostly based on our recurrence relation and Stich's analysis [1]. Thanks again for your insightful suggestion.
> > >
> > > [1] Sebastian U. Stich. Unified optimal analysis of the (stochastic) gradient method. arXiv preprint, 2019.

---

### Official Review · AnonReviewer2 · 2020-10-29
**Summary**

**Rating:** 7
**Confidence:** 4

**Review:**

This paper studies SSGD method to nonsmooth optimization problems with interpolation condition. It has proven that SSGD converges with rate O(1/epsilon) for convex problem and O(log(1/\epsilon)) for strongly-convex problems. What is more, it is proven that O(1/\epsilon) is optimal for the subgradient method in convex and interpolation setting.

Overall, I vote for accepting. The rate and lower bound results are solid theoretic results. It helps to explain the empirical observation that nonsmooth machine learning models are not necessarily more difficult to optimize than smooth models in practice.

Pros:

1. The results can be helpful to explain some observations in practice.

2. This would be a helpful result to investigate interpolation models, which has attracted attention recently.

Cons:

1. This paper only considers convex and strongly-convex problems, while over-parameterized models are usually nonconvex model, e.g., DNN.

2. The authors did not give empirical results to verify that (S)SGD do have similar performance in smooth and nonsmooth problems.

3. I have a minor concern regarding the reason of nonsmoothness. It seems that the authors blame the nonsmoothness in neural networks to the nonsmooth activation function. Even if the activation is smooth, the neural network is not L-smooth when w is unbounded, e.g., considering f(w)=w1*w_2*w_3.

Questions:
1. It is assumed that \ell(.) is 1-smooth. Fundamentally, is this crucial and why? I am not clear about this, especially considering that the inner function of the composite, i.e., h_i, is nonsmooth.

---

> ### Author Response · Authors · 2020-11-14
> **Thanks for your help feedback**
>
> Thanks for your helpful comments.
>
> **Q: Do we really need to assume that $\ell$ is 1-smooth?**
>
> A: We can just assume $\ell$ to be $\beta$-smooth for some $\beta > 0$. Our theorems still hold in general with minor changes in some constants depending on $\beta$. We assume $\ell$ to be 1-smooth in the paper is mostly for the following reasons: (1) simplicity and reader-friendly, (2) most commonly used loss functions are indeed 1-smooth e.g., square loss, negative entropy, and L2-hinge loss.
>
> **Q: The smoothness of neural networks?**
>
> A: Thanks for pointing out this question. Yes, we totally agree that neural networks with smooth activation can still be nonsmooth globally. But we may have smoothness if we are allowed to constrain the parameters of each layer to have bounded norm i.e., $\| W_i \| \leq B~ \forall i \in [n]$.
>
> Some numerical experiments are also included in the updated manuscript, please let us know if you have any questions regarding that section. Thanks again for your suggestions.

---

### Decision · Program_Chairs · 2021-01-07
**Final Decision**

**Decision:**

Accept (Poster)

**Comment:**

The paper proves new rates of convergence for stochastic subgradient under an interpolation condition. The analysis is rather simple but it produces better rates than previously known, which all reviewers agree is interesting. As pointed out by the reviewers, this work has the potential to help the community better understand optimization with over-parametrized neural networks (where convexity or other related assumptions play a role).

To the authors, please add a citation to Pegasos as requested by the reviewers.